# Usefulness of lung sound data collection using Skeeper SM-300® device: A pilot study

**Sue In Choi**[1☯], **Yujin Jeong**[2☯], **Won Jai Jung**[1], **Byung-Keun Kim**[1], **Sang Yeub Lee**[1], **Jungho Lee**[3], **JaeYong Kim**[3], **Won-Yang Cho**[3], **HyeSun Chang**[3], **Hyonggin An**[2], **Sanghoon Park**[4], **Eun Joo Lee**[1]*

1 Division of Respiratory, Allergy, and Critical Care Medicine, Korea University College of Medicine, Korea University Medical Center, Seoul, Republic of Korea, 2 Department of Biostatistics, Korea University College of Medicine, Seoul, Republic of Korea, 3 SmartSound Corporation, SmartSound Corporation, Seoul, Republic of Korea, 4 Department of Internal Medicine, Nokhyang Medical, Siheung-si, Gyeonggi-Do, Republic of Korea

☯ These authors equally contributed to this work.

* nanjung@korea.ac.kr

## Abstract

### Background

Digital stethoscope has been introduced to clinical practice and are developing in many aspects. Overwhelming pandemic requires non-contact healthcare support, and recent growth of artificial intelligence necessitates high-quality data collection.

### Objectives

We studied the usefulness of a novel, remotely applicable device Skeeper SM-300°.

### Methods

A single center, prospective, single-arm, investigator-blinded investigation was conducted. Sixty-four participants (normal individuals, stable/exacerbated COPD, stable/exacerbated bronchial asthma, pneumonia, ILD, and pleural effusion) had their lung sounds examined with a conventional stethoscope, while simultaneously recorded using the device, and classified into normal, crackle, wheezing and decreased sound. Three specialists (Raters) listened to the digitally recorded lung sounds and classified them. Non-inferiority of Skeeper SM-300° compared with conventional stethoscopy was examined. Audible acceptability of tidal breathing sounds compared with deep breathing sound, examinees' comfort with the new device, acceptability of digital lung sounds, and diagnostic accuracy after listening to the recorded lung sounds were also evaluated.

### Results

Among four breathing sounds, wheezing had the highest percentage of correct answers. Three Raters' judgment of classifying electronic lung sounds showed

**Data availability statement:** All relevant data are within the manuscript and its Supporting information files.

**Funding:** Research funding was raised by SIC from Smartsound Corporation. The funder provided support in the form of salaries for authors [JL, JYK, WYC, and HC], but did not have any additional role in the study design, data collection and analysis, decision to publish, or preparation of the manuscript.

**Competing interests:** I have read the journal's policy and the authors of this manuscript have the following competing interests: Research funding was raised by SIC from Smartsound Corporation. JL, JYK, WYC, and HC declare a potential conflict of interest due to holding stock/stock options in Smartsound Corporation. JL, JYK, WYC, and HC are employees of the company. All other authors declare no competing interests. And there is no competing interests relating to employment, consultancy, patents, products in development, or marketed products to this work. This does not alter our adherence to PLOS ONE policies on sharing data and materials.

consistent agreement with the conventional stethoscopic result. demonstrating statistically significant agreement between the conventional and electronic stethoscopy ($\kappa = 0.8337$, $\kappa = 0.8158$, and $\kappa = 0.8043$, respectively). Audible acceptability of tidal breathing sounds resulted in fair agreement, with a $\kappa$-value ranging from 0.5 to 0.8. A majority of patients expressed that they were fairly comfortable with being examined with the device. Diagnostic accuracy on listening to "deep breath" showed a nearly perfect agreement.

## Conclusion

We verified that the novel electronic stethoscope showed high credibility, yielded highly accurate results, and exhibited non-inferiority compared with conventional stethoscopy. Our investigation supports the usefulness of this novel digital auscultatory device as a credible substitute for conventional analogue stethoscopy.

## Introduction

Medicine is an ever-changing science. Not only discovery or diagnosis of human pathology, but pharmacologic or therapeutic approaches, prognostic prediction, and post-treatment management are evolving. Recently, medical devices for diagnosis and treatment of human ailment are rapidly being integrated with artificial intelligence (AI). Most of the adoption cases have been in imaging examination devices with some already in clinical use [1]. In contrast, there has been few instances regarding acoustic examination, e.g., auscultatory devices. The stethoscope is a cheap, light tool used by clinicians, but expertise and ample experience are required for proper use. If the AI is efficiently incorporated into a digital auscultatory device, vulnerable groups or areas will benefit from this novel technology. Moreover, pandemics such as COVID-19 commonly compels non-contact, non-face-to-face medical care, and a properly used digital auscultatory device will facilitate better medical service for patients who are strictly isolated for medical purpose.

Skeeper SM-300˚ is a novel digital, remotely usable auscultatory appliance from SmartSound Corporation (Fig 1). Since 2021, after being officially approved by the Korean Ministry of Food and Drug Safety, Korea University Medical Centre has been adopting Skeeper SM-300˚ for isolated subjects with COVID-19 for remote auscultation.

The aim of this study is to collect human lung sounds, both normal and diseased, captured with Skeeper SM-300˚, in order to accumulate sufficient data for the development of an AI analytic engine.

## Methods

### Participants enlistment

This investigation is a single center, prospective, single-arm, investigator-blinded investigation. Lung sounds were collected from voluntary participants. They were

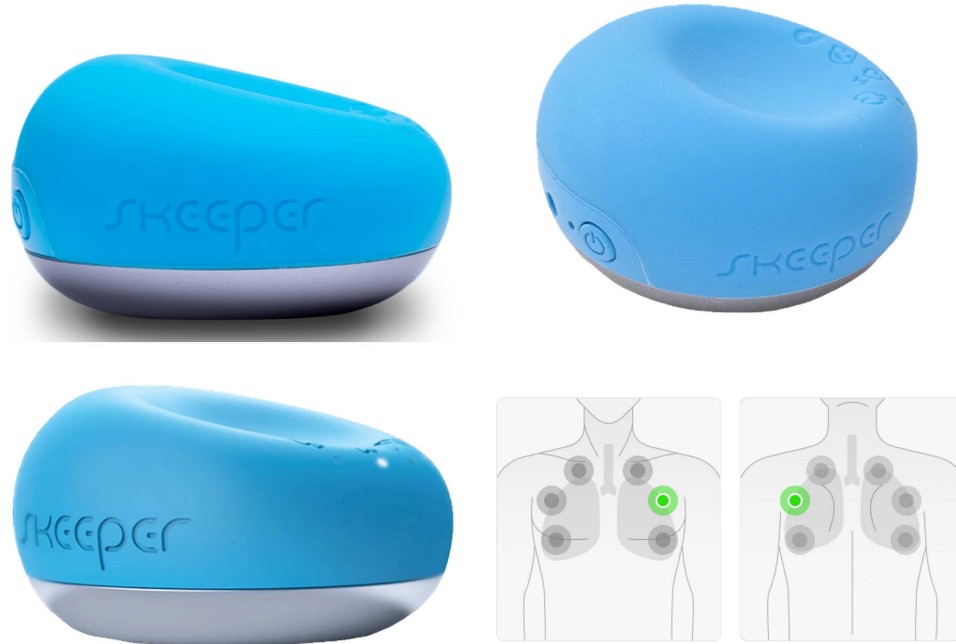

**Fig 1. Skeeper SM-300® device and foci for obtaining lung sounds.** An illustration of showing appoximate foci for obtaining lung sounds using Skeeper SM-300˚ device. Six foci of upper/middle/lower lung fields were chosen from the anterior chest and the back respectively, summing up to twelve foci from each participants.

recruited at the Department of Respiratory, Allergy and Critical Care Medicine, Korea University Medical Centre, Seoul, Korea from January 2023 to May 2024. Subjects consisted of healthy subjects and also disease-established patients with various lung diseases, and were classified as eight conditions. Conditions enlisted for this study were bronchial asthma (BA), chronic obstructive pulmonary disease (COPD), pneumonia, interstitial lung disease (ILD), and pleural effusion of any cause. Situations or conditions recorded from participants were eight in total as follows: 1) normal lung, 2) BA stable and 3) exacerbated, 4) COPD stable and 5) exacerbated, 6) pneumonia, 7) ILD, and 8) pleural effusion. Eight subjects respectively with these eight conditions, sixty-four participants in all, were participated for procuring lung sounds. As our study was a pilot clinical trial, the number of participants (n = 64) was decided without statistical review. This study was approved by the Korea University Medical Center Institutional Review Board (2021AN0582). Written consent was obtained from each examinee after sufficient explanation.

## Procurement of lung sound and other information

Each participant was checked for any symptoms or signs suggesting an ongoing acute illness (e.g., fever/chills, myalgia, cough/sputum, etc.). After verification, they were enrolled for lung sound procurement based on the following protocol.

The Skeeper SM-300˚ was used for obtaining lung sounds, mimicking conventional stethoscopy use. First, an expert pulmonologist with more than twenty years of experience (EJL, designated as Coordinator) performed conventional auscultation at twelve auscultatory spots – six from the anterior chest and six from the back – and marked them with an adhesive tape on participants' skin. Afterwards, the skin tape was removed and Skeeper SM-300˚ was gently placed on subjects' bare chest walls and backs, lung sounds were recorded for 1 minute during deep inspiration ("deep breath"),

followed by 1 minute recording of quiet, usual breathing ("tidal breath"). Participants were asked to smoothly and slowly inhale and exhale a "deep breath" and a "tidal breath" for 1 minute on each auscultation focus. Since the generation of crackles depends more on lung volume changes than on airflow, participants were guided to take slow and deep breaths in order to minimize flow turbulence and thus reduce the intensity of normal breath sounds [2]. The number of breaths was approximately ten and twenty times per minute, respectively. Auscultation and recording were performed on six foci on the anterior chest and back, respectively (Fig 1), totaling about twenty-four minutes of digital recording per individual. The obtained lung sounds were recorded on relevant digital terminals using a pre-installed application program, saving them as a digitalized audio computer file.

Information of participants' baseline characteristics, chest X-rays and if available, pulmonary function test (PFT) and chest computed tomography (CT) results were obtained during lung sound auscultation.

**Primary/secondary endpoints, and statistical analysis**

This study aimed at elucidating the following endpoints. Primary endpoint was to verify the non-inferiority of Skeeper SM-300® compared with conventional stethoscope use. Digitally recorded lung sounds (deep and tidal breaths) were classified as: 1) normal sound, 2) crackle, 3) wheezing, and 4) a decreased lung sound. A breathing sound was defined as a decreased lung sound if diminished breath sounds were present on one side relative to the opposite side when auscultated at the same anatomical level. Three expert clinicians, each with more than ten years of experience (WJJ, SIC and BGK, designated as Rater 1, 2, and 3 respectively), listened to the recorded "deep breath" lung sounds. The former two Raters were experienced pulmonologists, and the latter was an expert allergist. After listening to the electronically recorded lung sounds, the three Raters determined the lung sound from four choices. Consistency in choosing the type of breathing sound from the four classifications was required among the four investigators (Coordinator and Raters). A summarized flow diagram is shown in Fig 2.

If there was a discrepancy, the Coordinator listened to the recorded lung sound for verification. Due to inconsistencies in lung sound interpretation among investigators, the Coordinator additionally referred to a Mel-spectrogram (Fig 3), which was simultaneously obtained on digital recording. Briefly, mel-spectrogram is a colored, two-dimensional illustration in order to help enhancing human auditory perception [3]. The Coordinator therefore was able to confirm the type of breathing sound. Pairwise Cohen's Kappa coefficients were calculated to evaluate the agreement between each pair of the three Raters [4].

Concurrently, secondary endpoints were defined across four key areas: (1) the audible acceptability of "tidal breath" in comparison to "deep breath", (2) examinees' comfort level with the novel device versus a conventional stethoscope (assessed on a 2-step scale, the higher score, the more comfortable), (3) Raters' acceptance of lung sounds obtained with the new device compared to a conventional stethoscope (assessed on a 3-step scale, the higher score, the more acceptable), and (4) Raters' diagnostic accuracy for "deep breath" sounds.

(1) Audible acceptability of tidal breathing sounds was evaluated to determine if they are clear enough to provide useful medical information, comparable to deep breathing sounds. Repeated deep breathing can be uncomfortable and may cause dizziness in patients. Therefore, we evaluated the acceptability of tidal breaths compared to deep breaths using a digital device that can effectively amplify lung sounds. For the convenience of this study, two of the three Rater clinicians (WJJ and SIC) participated in this endpoint. They listened to the twelve pre-recorded "tidal breath" sounds and twelve "deep breath" sounds and judged whether the "tidal breath" sounds were identically audible to the "deep breath" sounds. Two clinicians' detections were compared using Cohen's Kappa point estimate with a 95% confidence interval (CI) for statistical verification.

(2) Examinees' comfort was subjectively assessed on a two-step scale (i.e., equal vs. more comfortable). The scales chosen by participants were analyzed using frequency analysis.

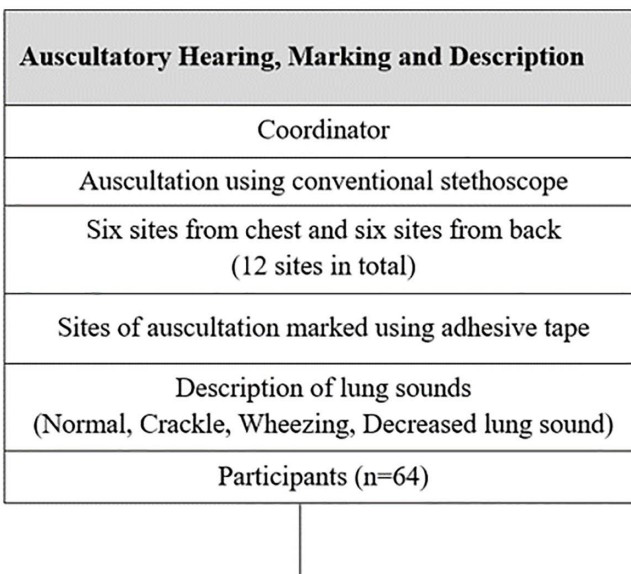

**Fig 2. Flow diagram of procuring and determining digital lung sounds using conventional and digital auscultation.**

(3) The third secondary endpoint, the acceptability of digital lung sound compared with conventional lung sounds by three Raters, was also evaluated and marked on a 3-step scale (i.e., negative, equal, and positive). The volume of lung sound and the presence of noise were evaluated. To compare reliability among the three Raters, intra-class correlation coefficient (ICC) was adopted, specifically Kendall's coefficient of concordance, used with a 95% CI, irrespective of site of the lung sound procurement site. We also categorized the acceptability result into three steps for simplicity: negative (originally 1), equal (originally 2, 3, 4), and positive (originally 5).

(4) Diagnostic accuracy after listening to the recorded lung sounds was evaluated as follows. For eight categorial conditions, the generalized kappa statistic method was used. The three Raters listened to recorded lung sounds of a total of 64 subjects and attempted to correctly classify the sounds among the eight given conditions. Fleiss' Kappa point estimate and Gwet's First-order Agreement Coefficient (AC1) point estimate were adopted, both with a 95% CI. Kappa coefficient and ICC values are interpreted as very good if they are ≥ 0.8, good if 0.6–0.8, moderate if 0.4–0.6, fair if they are 0.2–0.4, and poor agreement if ≤ 0.2 [5,6].

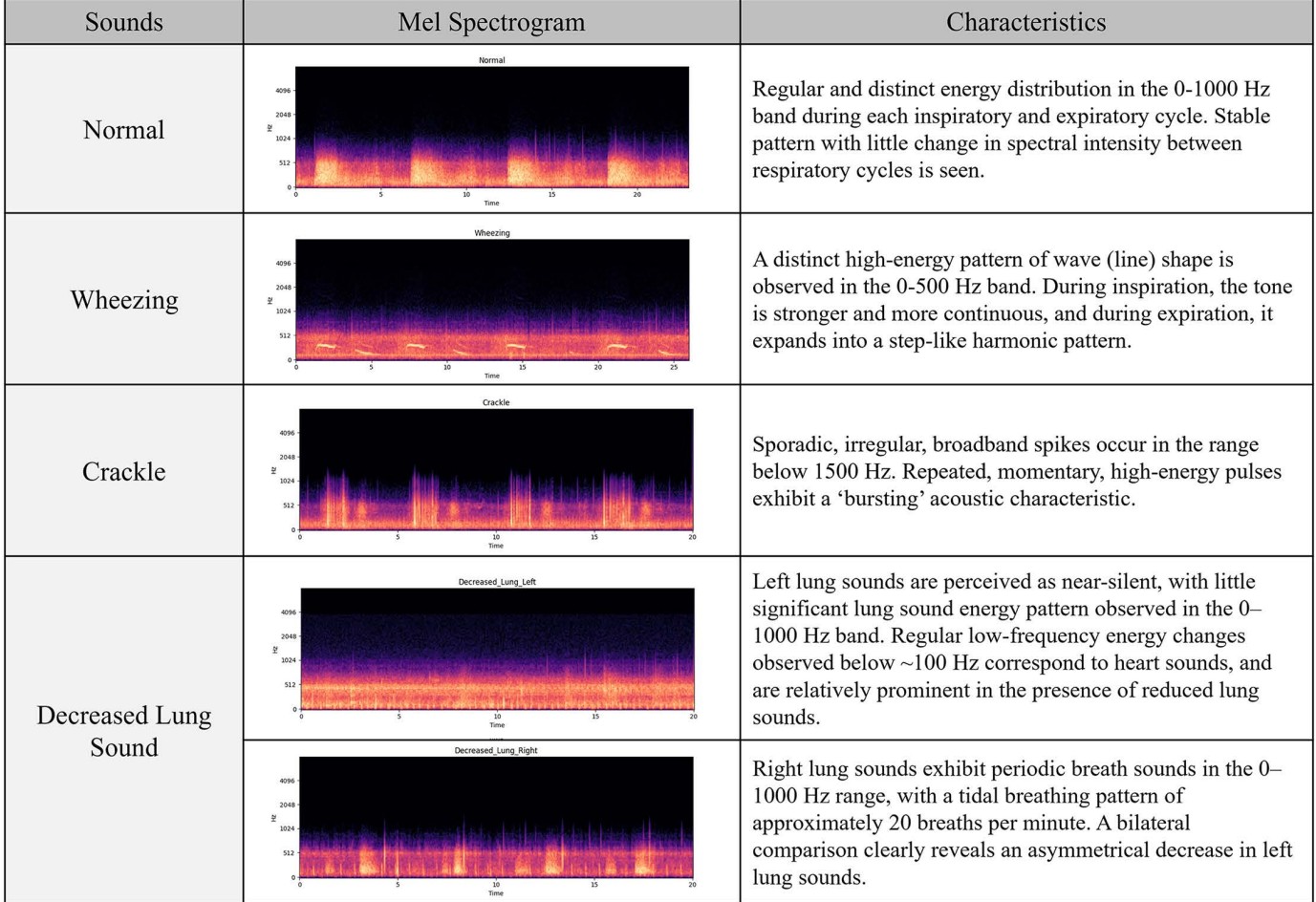

| Sounds | Mel Spectrogram | Characteristics |
|---|---|---|
| Normal | | Regular and distinct energy distribution in the 0-1000 Hz band during each inspiratory and expiratory cycle. Stable pattern with little change in spectral intensity between respiratory cycles is seen. |
| Wheezing | | A distinct high-energy pattern of wave (line) shape is observed in the 0-500 Hz band. During inspiration, the tone is stronger and more continuous, and during expiration, it expands into a step-like harmonic pattern. |
| Crackle | | Sporadic, irregular, broadband spikes occur in the range below 1500 Hz. Repeated, momentary, high-energy pulses exhibit a 'bursting' acoustic characteristic. |
| Decreased Lung Sound | | Left lung sounds are perceived as near-silent, with little significant lung sound energy pattern observed in the 0–1000 Hz band. Regular low-frequency energy changes observed below ~100 Hz correspond to heart sounds, and are relatively prominent in the presence of reduced lung sounds. |
| | | Right lung sounds exhibit periodic breath sounds in the 0–1000 Hz range, with a tidal breathing pattern of approximately 20 breaths per minute. A bilateral comparison clearly reveals an asymmetrical decrease in left lung sounds. |

**Fig 3. An exemplification of Mel-spectrograms, simultaneously obtained during lung sound recording, with a brief description of the characteristics of each lung sound.** A distinct difference in pictorial patterns is intuitively perceptible. For decreased lung sound, bilateral recordings (left and right) are presented to illustrate the asymmetric reduction in lung sound intensity. Regular low-frequency energy fluctuations observed below ~100 Hz on the affected side correspond to heart sounds, which become relatively prominent when lung sounds are diminished.

## Results

### Demographic information and lung sound

The study enrolled 64 participants in total. Their detailed demographic characteristics are summarized in Table 1. The normal lung group was the youngest, with a mean age of 41 years. Conversely, the exacerbated COPD group was the oldest, with a mean age of 81 years. All patients in the exacerbated COPD patients were male, while the exacerbated asthma group was predominantly female. Lung function was observed to be decreased in subjects within the COPD and ILD groups. PFT was not performed in participants of the Pneumonia and Pleural effusion groups. For the exacerbated asthma exacerbated COPD groups, lung sound recordings were taken after the symptom of acute stage of illness had abated, as significant dyspnea prevented earlier recording.

Breathing sounds were identified, collected and classified as normal, wheezing, crackle and decreased lung sound. Fig 2 and a digitally recorded audio data (Multimedia Index 1) exemplify lung sounds obtained from participants of our

**Table 1. Baseline characteristics of participants.**

| | Normal lung | Stable asthma | Exacerbated asthma | Stable COPD | Exacerbated COPD | Pneumonia | ILD | Pleural effusion |
|---|---|---|---|---|---|---|---|---|
| Age(years) | 40.9 | 61.3 | 64.3 | 76.1 | 81 | 76.5 | 74.3 | 64.4 |
| Female | 4 | 4 | 7 | 2 | 0 | 3 | 2 | 1 |
| PFT | | | | | | | | |
| FVC(L) | 4.1 | 3.2 | 2.9 | 3.1 | 2.5 | | 2.5 | |
| FVC (%pred) | 94.3 | 87.8 | 87.2 | 79.8 | 65.5 | | 64.4 | |
| FEV$_1$(L) | 3.4 | 2.4 | 1.9 | 1.7 | 1.2 | | 2.1 | |
| FEV$_1$(%pred) | 91.5 | 85.2 | 74.3 | 62.9 | 49.8 | | 81.1 | |
| FEV$_1$/FVC | 83.5 | 75.4 | 64.3 | 61.8 | 47 | | 85.1 | |

COPD: chronic obstructive pulmonary disease, ILD: interstitial lung disease, PFT: pulmonary function test, FVC: force volume capacity, FEV$_1$: forced expiratory volume on 1 second. PFT was not performed on participants of Pneumonia and Pleural effusion group.

study. Normal sound showed regular and stable pattern with little change in spectral intensity, as seen on the Mel-spectrogram. In contrast, decreased lung sounds showed almost no significant energy pattern. A distinct high-energy pattern of sharp line shape was observed in wheezing sounds. For crackle sounds, Mel-spectrogram revealed sporadic, irregular, and broadband spikes according to the inspiration/expiration cycle.

**A. Non-inferiority of Skeeper SM-300°.** Result of non-inferiority of novel stethoscopic device is summarized on Table 2. Among the four breathing sounds, wheezing had the highest percentage of correct answers (Rater 1: 93.7%, Rater 2: 91.8%, Rater 3: 98.0%), while crackle showed the lowest (80.2%, 80.9%, 77.0%, respectively). Notably, crackle and normal breath were occasionally difficult to identify; it was also tricky for Raters to differentiate normal sounds from lung sounds recorded up to 1 minute from bronchial asthma or COPD patients who had recovered from acute exacerbation, especially when faint, intermittent wheezing was still audible. Importantly, the three raters' classifications of electronic lung sounds were consistent with those obtained using a conventional stethoscope. Kappa coefficients of three clinicians were κ = 0.8337 (95% CI, 0.7964~0.8710, P < .0001), κ = 0.8158 (95% CI, 0.7767~0.8550, P < .0001), and κ = 0.8043 (95% CI, 0.7640~0.8445, P < .0001), respectively, showing statistically significant agreement between the conventional and electronic stethoscopy.

**B. Secondary endpoints. (1) Audible acceptability:** When evaluating the audible acceptability of recorded "breath" and "deep breath" sounds, two clinicians (Rater 1 and 2) showed fair agreement, with Kappa values ranging from 0.5 to 0.8. Table 3 shows each clinician's point estimate and 95% CI. The acronyms used refer to the focus of electronic auscultation on chest (front) or back (e.g., FL1 = front left 1, BR2 = back right 2). There was no prominent difference observed based upon the site of auscultation, whether it was the left or right lung, or the upper or lower lung field.

**(2) Examinees' comfort of new device compared with conventional stethoscope:** Patients found the device equally comfortable to be examined (79.69%) and more comfortable (20.31%).

**(3) Acceptability of lung sound compared with conventional stethoscope:** The result of simplified 3-step acceptability scale was as follows: negative (2.6%), equal (71.6%), and positive (25.8%). These findings demonstrate that the majority of the digitally record sounds from the participants' (n = 64) met Raters' standards. Gwet's AC1 point estimate revealed a point estimate of 0.5146 (CI 0.4211–0.6081), suggesting moderate agreement (Table 4).

**(4) Specialists' diagnostic accuracy on listening to "deep breath":** Fleiss's Kappa point estimate was 0.9047 (95% CI 0.8429–0.9665) and Gwet's AC1 point estimate was 0.9048 (95% CI 0.8426–0.9670). These two estimates demonstrated nearly perfect agreement (Table 5).

Table 2. Consistency of stethoscopic and digital lung sounds between the Coordinator and three Raters.

| Coordinator | Rater 1 | | | | |
|---|---|---|---|---|---|
| | Normal sound | Crackle | Wheezing | Decreased lung sound | Total |
| Normal sound | 429 | 15 | 11 | 4 | 459 |
| Crackle | 8 | 65 | 0 | 1 | 74 |
| Wheezing | 25 | 1 | 163 | 0 | 189 |
| Decreased lung sound | 5 | 0 | 0 | 30 | 35 |
| Total | 467 | 81 | 174 | 35 | 757 |
| Accuracy (%) | 91.9 | 80.2 | 93.7 | 85.7 | |

missing value number = 11

| Coordinator | Rater 2 | | | | |
|---|---|---|---|---|---|
| | Normal sound | Crackle | Wheezing | Decreased lung sound | Total |
| Normal sound | 435 | 13 | 15 | 3 | 466 |
| Crackle | 18 | 55 | 0 | 1 | 74 |
| Wheezing | 22 | 0 | 167 | 0 | 189 |
| Decreased lung sound | 5 | 0 | 0 | 30 | 35 |
| Total | 480 | 68 | 182 | 34 | 764 |
| Accuracy (%) | 90.6 | 80.9 | 91.8 | 88.2 | |

missing value number = 4

| Coordinator | Rater 3 | | | | |
|---|---|---|---|---|---|
| | Normal sound | Crackle | Wheezing | Decreased lung sound | Total |
| Normal sound | 428 | 19 | 3 | 4 | 454 |
| Crackle | 6 | 67 | 0 | 0 | 73 |
| Wheezing | 39 | 1 | 146 | 3 | 189 |
| Decreased lung sound | 6 | 0 | 0 | 29 | 35 |
| Total | 479 | 87 | 149 | 36 | 751 |
| Accuracy (%) | 89.4 | 77.0 | 98.0 | 80.6 | |

missing value number = 17

## Discussion

By our investigation, we verified that the electronic stethoscopic device Skeeper SM-300° offers high credibility, delivers highly accurate results, and is non-inferior to conventional stethoscopy. We also found that lung sounds were simultaneously identifiable through a visual pictogram (Mel-spectrogram), which allowed for better confirmation. Identifying wheeze proved excellent, with an over 90% consensus among Raters. However, crackle was occasionally mistaken for normal sounds due to background noise. Furthermore, lung sounds recorded up to one minute from patients in a convalescent phase sometimes revealed intermittent wheezes admixed with normal sounds, leading to discrepancies among Raters. There were cases where the classification of breath sounds decided by the Coordinator differed from the opinions of the majority of the Raters, so the Coordinator listened to the digitally recorded breath sounds again and checked the Mel-Spectrogram of the breath sounds, and in some cases, the Coordinator's original judgment was revised. This could have occurred because the Coordinator did not listen to the breath sounds with a traditional stethoscope at each site for a whole minute before classifying them as one of the four breath sounds. Considering the limited time for detailed auscultation in doctor's offices and clinics, Skeeper-recorded breath sounds of patients at home are less constrained by time and thus may provide important clues to clinicians.

Our investigation supports the usefulness of novel digital auscultatory device, showing that it can credibly substitute for conventional analogue stethoscopy. There have been several advancements in auscultatory devices for healthcare

**Table 3. Audible acceptability of the "tidal breath" as good as "deep breath". Rater 1 noticed more artifacts from skin friction, low lung sound level during shallower breathing, and/or loud ambient noise. The respective recordings were not counted for him.**

| Parameter | Missing values | Cohen's Kappa, κ | |
|---|---|---|---|
| | | Point estimate | 95% confidence limit |
| **Rater 1** | | | |
| FL1 | 4 | 0.6244 | 0.3974-0.8515 |
| FL2 | 9 | 0.6944 | 0.51-0.8789 |
| FL3 | 7 | 0.7932 | 0.65-0.9363 |
| FR1 | 2 | 0.7932 | 0.65-0.9363 |
| FR2 | 7 | 0.6904 | 0.5124-0.8684 |
| FR3 | 7 | 0.8324 | 0.7051-0.9598 |
| BL1 | 8 | 0.8324 | 0.7051-0.9598 |
| BL2 | 4 | 0.5732 | 0.3839-0.7624 |
| BL3 | 8 | 0.7385 | 0.5852-0.8917 |
| BR1 | 5 | 0.7385 | 0.5852-0.8917 |
| BR2 | 3 | 0.6361 | 0.4453-0.8269 |
| BR3 | 4 | 0.7307 | 0.583-0.8784 |
| **Parameter** | **Missing values** | **Cohen's Kappa, κ** | |
| | | Point estimate | 95% confidence limit |
| **Rater 2** | | | |
| FL1 | 0 | 0.7033 | 0.5025-0.9041 |
| FL2 | 0 | 0.5861 | 0.3911-0.7811 |
| FL3 | 1 | 0.5044 | 0.3166-0.6922 |
| FR1 | 0 | 0.5044 | 0.3166-0.6922 |
| FR2 | 0 | 0.5873 | 0.3983-0.7763 |
| FR3 | 0 | 0.5938 | 0.3992-0.7883 |
| BL1 | 0 | 0.6305 | 0.4323-0.8287 |
| BL2 | 0 | 0.5499 | 0.3643-0.7355 |
| BL3 | 2 | 0.7637 | 0.6232-0.9042 |
| BR1 | 2 | 0.5370 | 0.3149-0.7591 |
| BR2 | 0 | 0.6051 | 0.4184-0.7918 |
| BR3 | 0 | 0.5992 | 0.432-0.7663 |

(F; front, L; left, R; right, B; back).

**Table 4. Acceptability of lung sound by Raters compared with conventional stethoscope.**

| Parameter | Missing value | Ordinal response | | |
|---|---|---|---|---|
| | | Kendall's coefficient of concordance, W | Gwet's AC1 | |
| | | Point estimate | Point estimate | 95% confidence interval |
| Acceptability of lung sound by Raters | | 0.5509 | 0.5146 | 0.4211-0.6081 |

AC1, First-order agreement coefficient.

since the advent of traditional stethoscopes. The conventional stethoscope was invented by French doctor Laennec in 1818, originally in a form of monaural, hollow wooden tube [7]. After various updates and reforms, such as the introduction of flexible tubing and binaural auscultation, the traditional stethoscope has been adopted not only for cardiovascular or

**Table 5. Specialists' diagnostic accuracy on listening to "deep breath".**

| Parameter | Missing value | Nominal response | | | |
|---|---|---|---|---|---|
| | | Fleiss's Kappa | | Gwet's AC1 | |
| | | Point estimate | 95% confidence interval | Point estimate | 95% confidence interval |
| **Diagnostic accuracy on listening to "deep breath"** | | 0.9047 | 0.8429-0.9665 | 0.9048 | 0.8426-0.9670 |

AC1, First-order agreement coefficient.

pulmonary auscultation, but also for gastrointestinal examination. Since the introduction of modern acoustic stethoscopes by American cardiologist David Littmann in the 1960s and 1970s, the head of a stethoscope has included two parts: a smaller, open-chest bell and a larger, closed one (the diaphragm). While the bell is known to transmit lower frequencies better, the diaphragm is known to transmit higher frequencies better. However, according to a study by Nowak et al., no significant disproportions of low- and high-frequency contents are observed between the bell and the diaphragm [8]. On the other hand, some researchers explain that high-quality traditional stethoscopes can provide superior auscultation to clinicians while avoiding the electronic noise and handling artifacts common in digital devices [9].

During the COVID-19 pandemic, clinicians and healthcare professionals faced significant risks of exposure to highly contagious infectious agents. This highlights a persistent challenge, not to mention past outbreaks like the 2015 Middle East respiratory syndrome (MERS) in the Republic of Korea, the 2009 swine flu pandemic, or avian influenza. These lethal and transmissible infectious diseases commonly require strict quarantine management during treatment. Obtaining and listening to remotely recorded heart or lung sounds can effectively protect healthcare professionals from such highly contagious infections.

Another key utilization for this device could be at-home auscultatory monitoring of chronic lung illnesses such as bronchial asthma, COPD, or any other lung conditions. Subjects with chronic, commonly exacerbating illness could be educated on self-auscultation using the device. Their lung sounds would then be transferred to healthcare professionals for ongoing monitoring. Moreover, an onboard AI could be integrated into the appliance to interpret any pathologic lung sounds and alert the clinicians or patients for prompt, pre-emptive management. This approach could effectively reduce unnecessary emergency room visits or admissions. It could also help discern and rule out patients experiencing dyspnea due to psychological or panic-related causes, who might otherwise seek needless medical attention when notable pathological breathing lung sounds are present. A primary strength of digital auscultation technologies lies in their versatility for use outside traditional clinical settings, including patients' homes. This expanded reach facilitates a broader feedback loop from users, who serve as essential 'humans-in-the-loop' to drive the iterative advancement of machine learning models for chest sound analysis. Ultimately, the integration of diverse real-world data from these environments is crucial for enhancing the diagnostic precision and clinical reliability of AI-driven respiratory assessments [10].

Educational advantages are another strength of digital stethoscopes. Unlike conventional stethoscopes, which can only be used by a single examiner, digital stethoscopes can record auscultatory sounds acquired from subjects. This feature allows for the reproduction of recorded sounds, which is invaluable for teaching. "Hybrid" stethoscopes, featured with a shape of traditional stethoscope and equipped with a built-in digital recorder that can pick up auscultatory sounds, are already utilized in clinical practice. The Skeeper SM-300˚ is even more advanced and convenient tool compared to these existing stethoscopes, making it particularly useful for educating medical students and house residents.

The Skeeper SM-300˚ offers distinct advantages over existing digital stethoscopes, like 3M™ Littmann® CORE Digital Stethoscope, TytoCare or Sanolla. Notably, the Skeeper SM-300® is lighter and allows for easier handling of obtained sound as a digital file. Compared to the recent device, the 3M™ Littmann® CORE Digital Stethoscope (improved

piezoelectric sensor, 20–3,000 Hz, 40x amplification, Bluetooth, noise cancellation), the Skeeper SM-300® (MEMS, 20–4,000 Hz, 8 kHz sampling) has an advantage in a wider frequency range and reproduction of high-frequency breath sounds. This difference originates from the sensor technology each device uses [11]. The 3M™ Littmann® CORE Digital Stethoscope utilizes a piezoelectric sensor, which converts physical forces into electrical charges via the piezoelectric effect [12]. However, this sensor is notably less effective at collecting and recording higher frequency sounds. This limitation can result in suboptimal breathing sounds recordings, making the recorded file sound "unnatural" or differ from what is directly heard from the conventional stethoscope. In contrast, the new Skeeper SM-300° picks up patients' sound using a Micro Electro-Mechanical System (MEMS). MEMS technology covers a relatively wider frequency range without omitting higher Hz sounds. Consequently, this results in a more similar resonance and spatial impression compared to classical real-time stethoscopic auscultation, giving less of an impression of difference. Although the MEMS has some drawbacks such as inadvertent detection of irrelevant ambient sounds, this issue can be overcome with noise-cancelling technology in the near future. While TytoCare (features such as electronic filter-based, remote telemedicine, multi-examination) focuses on connectivity and Sanolla VoqX (features such as patented audio technology, infrasound, AI analysis) focuses on low-frequency analysis, Skeeper SM-300° is strong in clinical field breath sound recording with its MEMS-based wide frequency range (especially high frequencies).

Repetitive deep inspiration and expiration for auscultation can make examinees uncomfortable and even cause dizziness. Recording the auscultatory sounds can greatly ease patients' burden and physicians' effort during physical examinations. Another benefit is the ability to mark the site of auscultation during auscultation, which avoids confusion when gathering medical information and allow for convenient follow-up and comparison of an abnormal findings over the course of patient care.

This study was a pilot investigation, arbitrarily determining number of participants as 64. We performed a post-hoc power analysis based on the observed Cohen's Kappa coefficients. With a total sample size of 768 deep breathing digitalized audio computer file, the study achieved a statistical power of >80% (alpha = 0.05) to detect a significant agreement between the Coordinator and the Raters. This indicates that our sample size was more than sufficient to provide robust and reliable results, minimizing the risk of a Type II error.

Contrary to our assumption that tidal breathing sounds could fully substitute for the somewhat painstaking deep breathing sounds, they fell short of providing enough information. We hope that future technical advancement will allow more convenient and useful method for acquiring auscultatory lung sounds, simply through normal breathing without significant effort.

In conclusion, this investigation verified the usefulness and credibility of a novel device for acquiring human lung auscultatory sounds, facilitating non-contact, non-bedside physical examinations. The same device can also acquire cardiac sounds, and its utility evaluation is currently underway. We anticipate that in the near future, various devices will allow for the remote acquisition of other human body sounds, such as bowel sounds and voice phonation.

## Supporting information

**S1 File. Digitally recorded lung sound.** Normal lung sound recorded with Skeeper SM-300°. (MP4)

**S2 File. Digitally recorded lung sound.** Wheezing recorded with Skeeper SM-300°. (MP4)

**S3 File. Digitally recorded lung sound.** Crackle recorded with Skeeper SM-300°. (MP4)

**S4 File. Digitally recorded lung sound.** Decreased lung sound recorded with Skeeper SM-300°. (MP4)

**S5 File. Datasets used for statistical analysis.**
(XLSX)

## Author contributions

**Conceptualization:** Sue In Choi, Yujin Jeong, Byung-Keun Kim, Sanghoon Park, Eun Joo Lee.

**Data curation:** Sue In Choi, Yujin Jeong, Won Jai Jung, Byung-Keun Kim, Jungho Lee, JaeYong Kim, Won-Yang Cho, HyeSun Chang, Eun Joo Lee.

**Formal analysis:** Yujin Jeong, Byung-Keun Kim, Hyonggin An, Sanghoon Park, Eun Joo Lee.

**Funding acquisition:** Sue In Choi.

**Investigation:** Sue In Choi, Yujin Jeong, Won Jai Jung, Byung-Keun Kim, Eun Joo Lee.

**Methodology:** Sue In Choi, Yujin Jeong, Won Jai Jung, Byung-Keun Kim, Hyonggin An, Eun Joo Lee.

**Project administration:** Eun Joo Lee.

**Resources:** Jungho Lee, JaeYong Kim, Won-Yang Cho, HyeSun Chang, Eun Joo Lee.

**Software:** Yujin Jeong, Jungho Lee, JaeYong Kim, Won-Yang Cho, HyeSun Chang, Hyonggin An, Eun Joo Lee.

**Supervision:** Hyonggin An, Eun Joo Lee.

**Validation:** Sue In Choi, Yujin Jeong, Won Jai Jung, Byung-Keun Kim, Sang Yeub Lee, Jungho Lee, JaeYong Kim, Won-Yang Cho, HyeSun Chang, Hyonggin An, Sanghoon Park, Eun Joo Lee.

**Visualization:** Jungho Lee, JaeYong Kim, Won-Yang Cho, HyeSun Chang, Sanghoon Park, Eun Joo Lee.

**Writing – original draft:** Sanghoon Park, Eun Joo Lee.

**Writing – review & editing:** Sue In Choi, Yujin Jeong, Won Jai Jung, Byung-Keun Kim, Sang Yeub Lee, Jungho Lee, JaeYong Kim, Won-Yang Cho, HyeSun Chang, Hyonggin An, Sanghoon Park, Eun Joo Lee.

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
