## [Decision Letter · Decision Letter 0]

2 Feb 2026

PONE-D-25-42773Usefulness of Lung Sound Data Collection Using Skeeper SM-300 ®  DevicePLOS One

Dear Dr. Lee,

Thank you for submitting your manuscript to PLOS ONE. After careful consideration, we feel that it has merit but does not fully meet PLOS ONE’s publication criteria as it currently stands. Therefore, we invite you to submit a revised version of the manuscript that addresses the points raised during the review process.

Dear Author,

I would like to focus on the comments of Reviewers 1 and 3 during the manuscript revision process. Reviewer 2 thought that the paper should be rejected, and his comments, which require new research, are not relevant to the revision process.

Best regards,

Academic Editor

We look forward to receiving your revised manuscript.

Kind regards,

Damir Erceg, MD, PhD, Assoc. Prof.

Academic Editor

PLOS One

Journal Requirements:

Research funding was raised by SIC from Smartsound Corporation.

3. Thank you for stating the following in the Competing Interests/Financial Disclosure section:

I have read the journal's policy and the authors of this manuscript have the following competing interests: JL, JYK, WYC, HC, and EJL declare a potential conflict of interest due to holding stock/stock options in Smartsound Corporation, which is the funding source for this manuscript. JL, JYK, WYC, and HC are employees of the company. All other authors declare no competing interests.

We note that one or more of the authors are employed by a commercial company: Smartsound Corporation

4. Thank you for providing your underlying data as Supporting Information.

We note that the data set contains text or data that is not in English. Please note that PLOS is an English-language publisher, so we require data sets to be provided in English as well. Please upload an English-language version of your data set.

This will also allow us to determine if your data follows PLOS standards per our Data Availability policy here: https://journals.plos.org/plosone/s/data-availability

Reviewers' comments:

Reviewer's Responses to Questions

**Comments to the Author**

1. Is the manuscript technically sound, and do the data support the conclusions?

Reviewer #1: Partly

Reviewer #2: No

Reviewer #3: Partly

2. Has the statistical analysis been performed appropriately and rigorously? 

Reviewer #1: No

Reviewer #2: No

Reviewer #3: I Don't Know

3. Have the authors made all data underlying the findings in their manuscript fully available?

Reviewer #1: Yes

Reviewer #2: Yes

Reviewer #3: Yes

4. Is the manuscript presented in an intelligible fashion and written in standard English?

Reviewer #1: No

Reviewer #2: Yes

Reviewer #3: No

5. Review Comments to the Author

Reviewer #1: Dear Authors, actually it is not understandable to a reader what was the aim of your work. There are too many goals in the research. The sample size is not calculated especially because this is a non-inferiority study as you stated. It is also not known is the non-inferiority associated with all the goals that you mentioned. Your results show some discrepancy between two devices (up to 20%). It is very important how this would possibly influence the outcome of diagnosis and treatment in these subjects. Please correct these important issues.

Reviewer #2: I have significant concerns about this manuscript that prevent me from recommending publication in its current form:

Major Issues:

1. Insufficient sample size and study design: Only 8 participants per condition (n=64 total) is inadequate for a validation study. The authors acknowledge this is a "pilot" study without statistical power calculation, yet draw definitive conclusions about non-inferiority. This sample size is too small to support the broad claims made.

2. Severely limited literature review: Only 8 references is unacceptable for a scientific manuscript. The introduction lacks comprehensive discussion of existing digital stethoscope technology, previous validation studies, and AI applications in respiratory auscultation. The authors fail to adequately position their work within the existing body of knowledge.

3. Limited novelty: This is primarily a device validation study rather than a scientific contribution. Comparing a digital stethoscope to traditional auscultation has been done extensively. What new knowledge does this add to the field?

4. Conflicts of interest concerns: Multiple authors (JL, JYK, WYC, HC, EJL) hold stock in Smartsound Corporation, and several are employees. The study was funded by the device manufacturer. While disclosed, this raises concerns about bias and the independence of the research. The manuscript reads more like a technical validation report for regulatory purposes than independent scientific research.

5. Methodological limitations:

o Single-center study limits generalizability

o No comparison with other digital stethoscopes on the market

o Subjective classification system for lung sounds

o "Tidal breath" analysis showed only fair agreement, yet this limitation is downplayed

6. Overstated conclusions: The abstract and conclusion claim "non-inferiority" and "high credibility" based on limited data. The study shows reasonable agreement for wheezing (>90%) but poor performance for crackles (~80%), yet this is glossed over.

Minor Issues:

• Table 2 formatting could be improved

• Some grammatical inconsistencies throughout

• The mel-spectrogram discussion lacks depth

Recommendations: This manuscript requires major revisions including: expanded sample size with power analysis, comprehensive literature review (minimum 30-40 relevant references), comparison with competing devices, and more balanced discussion of limitations. In its current form, it does not meet the standards for publication in PLOS ONE.

Reviewer #3: The manuscript reports a diagnostic accuracy pilot study with a non-inferiority framework. Auscultatory findings with a traditional stethoscope are compared to reviews of these sounds as they were also recorded using Skeeter-SM-300, a "smart" Bluetooth electronic stethoscope that was introduced to the market in early 2023. The authors found highly accurate results and non-inferiority of Skeeter-SM-300 relative to auscultation with a conventional stethoscope.

Methods:

pg. 6, ln. 95 Participants were checked regarding ongoing acute illness, but it is not clear if or when those with positive findings were excluded.

pg. 6, ln. 113 A coordinator performed conventional auscultation over 6 anterior and 6 posterior sites. He/she marked these sites. Since this coordinator's auscultation served as the "ground truth", it needs to be explained that he/she documented the findings for each location and breath maneuver (deep vs tidal breathing), and that his/her marking of each location (ln. 115) served other coordinators (ln. 99) to immediately record with the Skeeter device at these sites to ideally capture the same sounds that he/she just heard.

pg. 7, ln. 100 The "deep breath" maneuvers of ~10 per minute followed by tidal breaths of ~20 per minute we likely taken at a higher-than-normal inspiratory flow rate. While a critical velocity of airflow is required to elicit airway wall flutter and wheezing, an increased airflow also increases normal lung sounds which then can mask the audibility of crackles. The details of the maneuver may be explained in the Methods section, and/or these influences should be mentioned in the Discussion.

pg. 7, ln, 101 Explain the rationale to record for 1 minute at 12 sites in 64 study subjects? Auscultation in clinical practice would be much shorter, perhaps two or three deeper breaths and over fewer than 12 sites.

pg. 7, ln. 103 One minute of recording at each of 12 sites of the chest should provide 12 minutes per study subject, not 24, unless there were separate "deep breath" and "tidal breath" recordings. Please clarify and consider a flow diagram for easier visualization of these steps of investigation.

pg. 7, ln. 116 Sounds were classified as normal, crackle, wheezing and decreased. In the medical conditions of enrolled patients, it can be expected that, depending on the sites of recording, each of these or even combinations can exist. How was this considered in subsequent analyses of agreement?

pg. 7, ln. 126 The coordinator who had established the "ground truth" (Reference or Gold Standard) by conventional auscultation confirmed the type of breathing sound in cases of disagreement between three raters. He/she additionally referred to Mel-spectrograms. Since he/she knew the "ground truth", there is the potential of bias when using a spectrogram for confirmation. Were there cases in which the spectrogram did not show what he/she had documented? Or did it perhaps show some finding that he/she had not documented but that one or more of the raters had heard? This information is needed to understand sensitivity, specificity and predictive values. Given the different frequency range that is covered by the Skeeter vs. traditional stethoscopes, this needs to be addressed.

pg. 7, ln. 132 The description of decreased lung sound in Fig. 2 claims that energy change over time is small. This is confusing since there are regularly spaced changes that are reflecting heart sounds which is confirmed on listening to the provided examples. On close audiovisual assessment, the logarithmic scale of the Mel-sonogram shows these heart sounds with most energy below 100 Hz but reaching up to ~600 Hz. Is this explained by the clinical status of the participant or perhaps by the acoustic characteristics of the Skeeter? Lung sounds of very low intensity can be distinguished at a rate of around 20 per minute, indicating tidal breathing. The presentation of this spectral plot only makes sense when also showing the recording over the corresponding other side.

pg. 8, ln. 148 Clarify whether the 2 selected raters listened to "the" pre-recorded sounds of 12 tidal and 12 deep breaths in all recordings, i.e. from all 12 sites in all 8 subjects. What does-"pre-"recorded mean in this context?

pg. 8., ln. 153 The secondary outcome of participants comfort is overly complicated. Why five levels of discomfort/comfort? This aspect of the study could be shortened since the participants found the Skeeter either equally or more comfortable than a traditional stethoscope.

pg. 8, ln. 159 Here, the acceptability of digital lung sound also asks for a distinction of "better and considerable acceptable" vs "very acceptable". Without definition of such subtle grading, acceptance by subjects and raters should simply be negative, equal and positive.

pg. 9, ln. 165 When 3 of 64 participants' recordings were arbitrarily (?randomly) chosen to be tested for diagnostic accuracy, was it possible that all 3 came from one of the 8 participants' categories, e.g. all could have had wheezing in acute asthma or crackles in ILD? If not randomly chosen, who was the arbitrator and what were the selection criteria?

Results:

pg. 11, ln. 208 "Crucially, the three Raters’ judgments in classifying electronic lung sounds consistently coincided with the conventional stethoscopic result" should better read "Importantly, the three raters’ classifications of electronic lung sounds were consistent with those obtained using a conventional stethoscope".

Tbl. 3 What explains the significantly higher number of missing values for Rater 1 (68) vs Rater 2 (5)?

Tbl. 4 I suggest describing these findings in the text and omit this table.

Discussion:

pg. 19, ln. 267 Remove “i.e., discontinuous”. Regarding lung sounds, “discontinuous” refers to crackles.

The discussion should present information on the acoustic performance of traditional stethoscopes since the goal of the study was to show non-inferiority. More recent studies are "Acoustical properties of amplified and unamplified stethoscopes when examining typical body sounds" by JA Dunnington in 2017 (doctoral thesis, publicly available at https://digitalcommons.latech.edu/dissertations) and “Acoustic characterization of stethoscopes using auscultation sounds as test signals” by LJ Novak (J Acoust Soc Am 2017; 141(3):1940. doi: 10.1121/1.4978524 ).

The comparison of Skeeter-SM-300 with the Littman Model 3200 stethoscope is not informative. Littman replaced that electronic stethoscope by the CORE model five years ago. The Littman CORE stethoscope has higher amplification, use of Bluetooth, active noise cancellation etc. Comparison with other commercially available stethoscopes might also mention more recently introduced devices, e.g. from TytoCare (https://www.tytocare.com/how-does-tytocare-work/) or Sanolla (https://sanolla.com/product/shop-voqx/) that take different approaches.

As the authors point out, a particular strength of the Skeeter is its usefulness away from the hospital or doctors’ office, i.e. in the homes of patients. This offers a much broader feedback from users who are still needed as the "humans-in-the-loop" to advance the machine learning of chest auscultation (https://doi.org/10.1016/j.chpulm.2024.100079).

6. PLOS authors have the option to publish the peer review history of their article (what does this mean?). If published, this will include your full peer review and any attached files.

Reviewer #1: No

Reviewer #2: No

Reviewer #3: **Yes:** Hans Pasterkamp

---

## [Author Response · Author response to Decision Letter 1]

6 Mar 2026

Dear Author,

I would like to focus on the comments of Reviewers 1 and 3 during the manuscript revision process. Reviewer 2 thought that the paper should be rejected, and his comments, which require new research, are not relevant to the revision process.

Best regards, Academic Editor

Review Comments to the Author

Reviewer #1: Dear Authors, actually it is not understandable to a reader what was the aim of your work. There are too many goals in the research. The sample size is not calculated especially because this is a non-inferiority study as you stated. It is also not known is the non-inferiority associated with all the goals that you mentioned. Your results show some discrepancy between two devices (up to 20%). It is very important how this would possibly influence the outcome of diagnosis and treatment in these subjects. Please correct these important issues.

We appreciate the reviewer’s comment regarding the clarity of our research objectives. Our study is a pilot clinical study, therefore number of participants (n=64) was decided without statistical review. We performed a post-hoc power analysis based on the observed Cohen’s Kappa coefficients. With a total sample size of over 750 lung sound files, the study achieved a statistical power of >80% (alpha=0.05) to detect a significant agreement between the Coordinator and the Raters. This indicates that our sample size was more than sufficient to provide robust and reliable results, minimizing the risk of a Type II error. We added several sentences to our revised manuscript. Please refer to the “Discussion” section of our manuscript. Thank you for your meaningful opinion for clarifying our manuscript.

Reviewer #3: The manuscript reports a diagnostic accuracy pilot study with a non-inferiority framework. Auscultatory findings with a traditional stethoscope are compared to reviews of these sounds as they were also recorded using Skeeter-SM-300, a "smart" Bluetooth electronic stethoscope that was introduced to the market in early 2023. The authors found highly accurate results and non-inferiority of Skeeter-SM-300 relative to auscultation with a conventional stethoscope.

We sincerely thank Reviewer 3 for the thorough and insightful review. Your comments have helped us significantly improve the clarity of our methodology and the depth of our discussion. We have addressed each point as follows:

Methods:

pg. 6, ln. 95 Participants were checked regarding ongoing acute illness, but it is not clear if or when those with positive findings were excluded.

We verified participants’ acute illness on registering patients for study enrollment. On our submitted and reviewed manuscript, we already described as follows: “Each participant was checked for any symptoms or signs suggesting an ongoing acute illness (e.g., fever/chills, myalgia, cough/sputum, etc.). After verification, they were enrolled for lung sound procurement based on the following protocol.” Please refer to the subsection Procurement of Lung Sound and Other Information of Methods section.

pg. 6, ln. 113 A coordinator performed conventional auscultation over 6 anterior and 6 posterior sites. He/she marked these sites. Since this coordinator's auscultation served as the "ground truth", it needs to be explained that he/she documented the findings for each location and breath maneuver (deep vs tidal breathing), and that his/her marking of each location (ln. 115) served other coordinators (ln. 99) to immediately record with the Skeeter device at these sites to ideally capture the same sounds that he/she just heard.

Thank you for your kind comment. A coordinator with over 20 years of experience as a respiratory physician first performed auscultation using a conventional, analog stethoscope at six designated sites on the anterior chest and six sites on the back. After auscultation at each site, the locations were immediately marked using skin tape and the auscultation results were also simultaneously described. Immediately afterwards, the skin tape was removed and lung sound recorded with Skeeper SM-300®, and lung sounds were recorded for 1 minute during deep inspiration (considering that deep inspiration can cause effort and fatigue in the patient), followed by 1 minute of tidal breathing. For future readers’ convenience, we newly constructed a figure (Figure 2) regarding the process of data procurement. Please refer to our revised manuscript.

pg. 7, ln. 100 The "deep breath" maneuvers of ~10 per minute followed by tidal breaths of ~20 per minute we likely taken at a higher-than-normal inspiratory flow rate. While a critical velocity of airflow is required to elicit airway wall flutter and wheezing, an increased airflow also increases normal lung sounds which then can mask the audibility of crackles. The details of the maneuver may be explained in the Methods section, and/or these influences should be mentioned in the Discussion.

Thank you for your useful comment. We made a comment in our manuscript regarding the “deep breath” to smoothly and slowly inhale, in order to avoid masking the audibility of crackles by normal lung sound. We also read a useful article regarding this matter and added to our bibliographic list (Kiyokawa H, Greenberg M, Shirota K, Pasterkamp H. "Auditory detection of simulated crackles in breath sounds." Chest. 2001 Jun;119(6):1886-92), and wrote several sentences on the Method section. Once again, thank you for your meaningful opinion for clarifying our manuscript.

pg. 7, ln, 101 Explain the rationale to record for 1 minute at 12 sites in 64 study subjects? Auscultation in clinical practice would be much shorter, perhaps two or three deeper breaths and over fewer than 12 sites.

We completely agree. In clinical practice, auscultating a few breath sounds at fewer than 12 sites may be sufficient. While lung sounds should ideally be listened to from both the upper, middle, and lower sides of the chest and back, this is often not practical in real world. Nevertheless, this study aimed to verify the usefulness of collecting lung sounds with a new device by collecting as many lung sounds as possible from each subject, eliminating any artifacts that inevitably occur, and capturing accurate lung sounds. For the convenience of the participants, we recorded deep breathing sounds for 1 minute, followed immediately by tidal breathing sounds at the same location.

pg. 7, ln. 103 One minute of recording at each of 12 sites of the chest should provide 12 minutes per study subject, not 24, unless there were separate "deep breath" and "tidal breath" recordings. Please clarify and consider a flow diagram for easier visualization of these steps of investigation.

We completely agree with your opinion regarding the steps of investigation, and we have developed a flow diagram as previously commented. Please refer to the newly made Figure 2. We hope this new figure will help our future readers understand our study more clearly.

pg. 7, ln. 116 Sounds were classified as normal, crackle, wheezing and decreased. In the medical conditions of enrolled patients, it can be expected that, depending on the sites of recording, each of these or even combinations can exist. How was this considered in subsequent analyses of agreement?

As the reviewer noted, during recording, the lack of contact between the Skeeper device and the skin often made it difficult to distinguish between normal sounds with added artifacts and true crackles. Before each Rater evaluated the digitally recorded lung sounds, the Coordinator and Raters altogether gathered to listen to several sample recorded lung sounds and established a baseline, or "ground truth," after which each Rater evaluated independently in an independent room. Lung sounds were then classified based on the predominant sound heard during the first minute.

pg. 7, ln. 126 The coordinator who had established the "ground truth" (Reference or Gold Standard) by conventional auscultation confirmed the type of breathing sound in cases of disagreement between three raters. He/she additionally referred to Mel-spectrograms. Since he/she knew the "ground truth", there is the potential of bias when using a spectrogram for confirmation. Were there cases in which the spectrogram did not show what he/she had documented? Or did it perhaps show some finding that he/she had not documented but that one or more of the raters had heard? This information is needed to understand sensitivity, specificity and predictive values. Given the different frequency range that is covered by the Skeeter vs. traditional stethoscopes, this needs to be addressed.

Inter-Rater discrepancies occurred primarily in the evaluation of crackles and normal sounds with added artifacts due to skin friction. In these cases, Mel- spectrograms were also checked. Meanwhile, wheezing not audible with traditional stethoscopes was recorded, or mild wheezing audible with traditional stethoscopes was lost during relatively long recordings. In these cases, the Coordinator had to directly listen to the lung sounds recorded with the Skeeper and revise the evaluation. However, there were no cases of discrepancies between the Mel-spectrogram and the presence or absence of wheezing. Furthermore, breathing sounds from the Skeeper device is very similar to that of a traditional stethoscope in that it records and reproduces a wide frequency range and high-frequency breath sounds. This content has been added to the discussion. Please review our revised paper. Thank you for pointing out the areas that researchers have struggled with.

pg. 7, ln. 132 The description of decreased lung sound in Fig. 2 claims that energy change over time is small. This is confusing since there are regularly spaced changes that are reflecting heart sounds which is confirmed on listening to the provided examples. On close audiovisual assessment, the logarithmic scale of the Mel-sonogram shows these heart sounds with most energy below 100 Hz but reaching up to ~600 Hz. Is this explained by the clinical status of the participant or perhaps by the acoustic characteristics of the Skeeter? Lung sounds of very low intensity can be distinguished at a rate of around 20 per minute, indicating tidal breathing. The presentation of this spectral plot only makes sense when also showing the recording over the corresponding other side.

The description of decreased lung sounds in Fig. 2 as "small energy change over time" was intended to emphasize that lung sounds are generally weak and show small temporal variations throughout the respiratory cycle. It did not apply to all acoustic components, including heart sounds. The revised manuscript (the legend for Figure 3) clarifies that this description is limited to lung sounds. The regular low-frequency energy changes observed in the presented Mel spectrogram correspond to heart sounds, as pointed out by the reviewer. In this example, normal heart sounds are relatively more pronounced in clinical situations with decreased lung sounds.

To answer the reviewer's question, "Is this due to the participant's clinical condition or the acoustic characteristics of the Skeeper?", the Skeeper SM-300 is a wideband digital stethoscope capable of measuring heart and lung sounds in the range of approximately 20–1500 Hz. The pattern is not an artifact of the device's acoustic characteristics, but rather a result of the participant's clinical condition showing decreased lung sounds. Furthermore, as the reviewer mentioned, in the example, very faint lung sounds were discernible at approximately 20 beats per minute, which is consistent with the normal respiratory rate of tidal breathing. In response to the reviewer's suggestion for a bilateral spectral comparison, the revised Fig. 3 presents the contralateral recording of the decreased lung sound example. Figure 3 emphasizes this bilateral comparison and more clearly describes the representative spectral characteristics of each category.

pg. 8, ln. 148 Clarify whether the 2 selected raters listened to "the" pre-recorded sounds of 12 tidal and 12 deep breaths in all recordings, i.e. from all 12 sites in all 8 subjects. What does-"pre-"recorded mean in this context?

We are terribly sorry for this error, and the word was changed into “recorded”. Thank you for your comment.

pg. 8., ln. 153 The secondary outcome of participants comfort is overly complicated. Why five levels of discomfort/comfort? This aspect of the study could be shortened since the participants found the Skeeter either equally or more comfortable than a traditional stethoscope.

Based on your opinion, we simplified the comfortability into two steps (equal or more comfortable). Thank you for your meticulous comment. Please refer to the changes of our manuscript in Method and Result section (we also deleted Table 4 as recommended).

pg. 8, ln. 159 Here, the acceptability of digital lung sound also asks for a distinction of "better and considerable acceptable" vs "very acceptable". Without definition of such subtle grading, acceptance by subjects and raters should simply be negative, equal and positive.

Based on your opinion, we simplified it into 3 steps: negative (originally 1), equal (originally 2, 3, 4), and positive (originally 5). We newly described this matter on the Method and Result sections of our manuscript. However, while the comfort survey of the participants described above was limited to checking the frequency, we inform you that the correlation coefficient statistics results using the data from the survey, which divided the acceptability into five levels for three Raters, have been used as is for the convenience of readers.

pg. 9, ln. 165 When 3 of 64 participants' recordings were arbitrarily (?randomly) chosen to be tested for diagnostic accuracy, was it possible that all 3 came from one of the 8 participants' categories, e.g. all could have had wheezing in acute asthma or crackles in ILD? If not randomly chosen, who was the arbitrator and what were the selection criteria?

In this analysis, the diagnostic accuracy was evaluated using lung sounds of a total of 64 subjects. Three Raters listened to the lung sounds of each subject and classified them into one of eight predefined diagnostic categorial conditions. The Fleiss kappa point estimate and Gwet's ACI value were checked to confirm the agreement between the three Raters and the agreement with the correct answer. We apologize for the confusion caused by the incorrect description as if the study was conducted by selecting only three patients, and we have re-described it to match the study that was conducted.

Results:

pg. 11, ln. 208 "Crucially, the three Raters’ judgments in classifying electronic lung sounds consistently coincided with the conventional stethoscopic result" should better read "Importantly, the three raters’ classifications of electronic lung sounds were consistent with those obtained using a conventional stethoscope".

We modified the sentence as recommended. Thank you for your kind advice.

Tbl. 3 What explains the significantly higher number of missing values for Rater 1 (68) vs Rater 2 (5)?

According to the Rater 1, the recorded sound file contained many artifacts such as skin friction sounds making it difficult to classify lung sounds, or the surrounding noise was loud, or the lung sounds were recorded too quietly to make a judgment on lung sound classification, so they were put on hold. In addition, the space where the Raters listened to the lung sounds was not a perfectly sound-insulated chamber. Therefore, the difference in the listening environment for each Rater may have affected the missing values. We would like to reiterate that the Coordinator did not influence the evaluation results of each Rater and respected the opinions of the Rater while conducting the study. On the other hand, during deep breathing, the Rater had few missing values without much differen

---

## [Decision Letter · Decision Letter 1]

25 Mar 2026

PONE-D-25-42773R1

Usefulness of Lung Sound Data Collection Using Skeeper SM-300 ®  Device

PLOS One

Dear Dr. Lee,

Thank you for submitting your manuscript to PLOS ONE. After careful consideration, we feel that it has merit but does not fully meet PLOS ONE’s publication criteria as it currently stands. Therefore, we invite you to submit a revised version of the manuscript that addresses the points raised during the review process.

A letter that responds to each point raised by the academic editor and reviewer(s). You should upload this letter as a separate file labeled 'Response to Reviewers'.A marked-up copy of your manuscript that highlights changes made to the original version. You should upload this as a separate file labeled 'Revised Manuscript with Track Changes'.An unmarked version of your revised paper without tracked changes. You should upload this as a separate file labeled 'Manuscript'

We look forward to receiving your revised manuscript.

Kind regards,

Damir Erceg, MD, PhD, Assoc. Prof.

Academic Editor

PLOS One

Journal Requirements:

Reviewer's Responses to Questions

**Comments to the Author**

1. If the authors have adequately addressed your comments raised in a previous round of review and you feel that this manuscript is now acceptable for publication, you may indicate that here to bypass the “Comments to the Author” section, enter your conflict of interest statement in the “Confidential to Editor” section, and submit your "Accept" recommendation.

Reviewer #1: All comments have been addressed

Reviewer #3: All comments have been addressed

2. Is the manuscript technically sound, and do the data support the conclusions?

Reviewer #1: Partly

Reviewer #3: Yes

3. Has the statistical analysis been performed appropriately and rigorously?

Reviewer #1: Yes

Reviewer #3: I Don't Know

4. Have the authors made all data underlying the findings in their manuscript fully available?

Reviewer #1: Yes

Reviewer #3: Yes

5. Is the manuscript presented in an intelligible fashion and written in standard English?

Reviewer #1: Yes

Reviewer #3: Yes

6. Review Comments to the Author

Reviewer #1: Please as this is the pilot study, you should mention it in the title of the manuscript: Usefulness of Lung Sound Data Collection Using Skeeper SM-300 ® Device: A pilot study

Reviewer #3: I appreciate the changes you made. There are a few remaining issues that you can also see on an uploaded PDF file where I have used strike-out and highlighting for suggested modifications.

Remove detail of your initial 5-level scale of audible acceptability (see page 9, ln 166-174) and references to this elsewhere. The change to just three categories (negative, equal, positive) is logical and will not only appeal to this reviewer but to all readers of your manuscript.

Shorten text on pg. 21, ln 280-291

There were cases where the classification of breath sounds decided by the Coordinator differed from the opinions of the majority of the Raters, so the Coordinator listened to the digitally recorded breath sounds again and checked the Mel-Spectrogram of the breath sounds, and in some cases, the Coordinator's original judgment was revised. This could have occurred because the Coordinator did not listen to the breath sounds with a traditional stethoscope at each site for a whole minute before classifying them as one of the four breath sounds. Considering the limited time for detailed auscultation in doctor’s offices and clinics, Skeeper-recorded breath sounds of patients at home are less constrained by time and thus may provide important clues to clinicians.

Legend for Table 3

Your explanation for the large number of missing values for Rater 1 that you provide to me in your answer has to be added so that all readers who might wonder about possible reasons can understand, e.g., Rater 1 noticed more artifacts from skin friction, low lung sound level during shallower breathing, and/or loud ambient noise. The respective recordings were not counted for him.

7. PLOS authors have the option to publish the peer review history of their article (what does this mean?). If published, this will include your full peer review and any attached files

**Do you want your identity to be public for this peer review?** For information about this choice, including consent withdrawal, please see our Privacy Policy.

Reviewer #1: No

Reviewer #3: No

---

## [Author Response · Author response to Decision Letter 2]

30 Mar 2026

Reviewer #1: Please as this is the pilot study, you should mention it in the title of the manuscript: Usefulness of Lung Sound Data Collection Using Skeeper SM-300 ® Device: A pilot study

Thank you for your kind comment and recommendation. We modified the title as requested.

Reviewer #3: I appreciate the changes you made. There are a few remaining issues that you can also see on an uploaded PDF file where I have used strike-out and highlighting for suggested modifications.

Remove detail of your initial 5-level scale of audible acceptability (see page 9, ln 166-174) and references to this elsewhere. The change to just three categories (negative, equal, positive) is logical and will not only appeal to this reviewer but to all readers of your manuscript.

Thank you for your kind comment and recommendation. We are terribly sorry for this mistake we made and corrected our error in our manuscript as recommended. Please refer to the revised manuscript (we made a green-colored mark on the requested modification on page 8 and 9.

Shorten text on pg. 21, ln 280-291

There were cases where the classification of breath sounds decided by the Coordinator differed from the opinions of the majority of the Raters, so the Coordinator listened to the digitally recorded breath sounds again and checked the Mel-Spectrogram of the breath sounds, and in some cases, the Coordinator's original judgment was revised. This could have occurred because the Coordinator did not listen to the breath sounds with a traditional stethoscope at each site for a whole minute before classifying them as one of the four breath sounds. Considering the limited time for detailed auscultation in doctor’s offices and clinics, Skeeper-recorded breath sounds of patients at home are less constrained by time and thus may provide important clues to clinicians.

We appreciate your meticulous editing of our manuscript and modified several sentences identical to the PDF file you have sent us. Please refer to the change with green-colored mark in our manuscript on page 21.

Legend for Table 3

Your explanation for the large number of missing values for Rater 1 that you provide to me in your answer has to be added so that all readers who might wonder about possible reasons can understand, e.g., Rater 1 noticed more artifacts from skin friction, low lung sound level during shallower breathing, and/or loud ambient noise. The respective recordings were not counted for him.

Thank you for your comment. We added sentences in your PDF file to the legend of Table 3.

---

## [Decision Letter · Decision Letter 2]

20 Apr 2026

Usefulness of Lung Sound Data Collection Using Skeeper SM-300 ®  Device: a Pilot Study

PONE-D-25-42773R2

Dear Dr. Lee,

We’re pleased to inform you that your manuscript has been judged scientifically suitable for publication and will be formally accepted for publication once it meets all outstanding technical requirements.

Kind regards,

Damir Erceg, MD, PhD, Assoc. Prof.

Academic Editor

PLOS One

Additional Editor Comments (optional):

Reviewers' comments:

Reviewer's Responses to Questions

**Comments to the Author**

1. If the authors have adequately addressed your comments raised in a previous round of review and you feel that this manuscript is now acceptable for publication, you may indicate that here to bypass the “Comments to the Author” section, enter your conflict of interest statement in the “Confidential to Editor” section, and submit your "Accept" recommendation.

Reviewer #1: All comments have been addressed

Reviewer #3: All comments have been addressed

2. Is the manuscript technically sound, and do the data support the conclusions?

Reviewer #1: Yes

Reviewer #3: (No Response)

3. Has the statistical analysis been performed appropriately and rigorously? 

Reviewer #1: Yes

Reviewer #3: (No Response)

4. Have the authors made all data underlying the findings in their manuscript fully available?

Reviewer #1: Yes

Reviewer #3: (No Response)

5. Is the manuscript presented in an intelligible fashion and written in standard English?

Reviewer #1: Yes

Reviewer #3: (No Response)

6. Review Comments to the Author

Reviewer #1: Dear Authors, you have made all suggested changes that I have made and now your manuscript is acceptable for publication.

Reviewer #3: (No Response)

7. PLOS authors have the option to publish the peer review history of their article (what does this mean?). If published, this will include your full peer review and any attached files.

Reviewer #1: No

Reviewer #3: No

---

## [Editor Report · Acceptance letter]

PONE-D-25-42773R2

PLOS One

Dear Dr. Lee,

I'm pleased to inform you that your manuscript has been deemed suitable for publication in PLOS One. Congratulations! Your manuscript is now being handed over to our production team.

Kind regards,

on behalf of

Dr. Damir Erceg

Academic Editor

PLOS One